# Bibliometric Analysis of Methods and Tools for Drought Monitoring and Prediction in Africa

**Omolola M. Adisa [1,2,]*, Muthoni Masinde [1], Joel O. Botai [1,2] and Christina M. Botai [2]**

[1] Department of Information Technology, Central University of Technology, Free State, Private Bag X200539, Bloemfontein 9300, South Africa; emasinde@cut.ac.za (M.M.); joel.botai@weathersa.co.za (J.O.B.)

[2] South African Weather Service, Private Bag X097, Pretoria 0001, South Africa; christina.botai@weathersa.co.za

* Correspondence: lolaadisa@yahoo.com; Tel.: +27-848491170

**Abstract:** The African continent has a long history of rainfall fluctuations of varying duration and intensities. This has led to varying degrees of drought conditions, triggering research interest across the continent. The research presented here is a bibliometric analysis of scientific articles on drought monitoring and prediction published in Africa. Scientific data analysis was carried out based on bibliometric mapping techniques applied to 332 scientific publications (1980 to 2020) retrieved from the Web of Science (WoS) and Scopus databases. In addition, time series of Standardized Precipitation Evapotranspiration Index for the previous 6 months (SPEI-6) over six regions in the continent was analysed giving the relative comparison of drought occurrences to the annual distribution of the scientific publications. The results revealed that agricultural and hydrological drought studies contributed about 75% of the total publications, while the remaining 25% was shared among socioeconomic and meteorological studies. Countries in the southern, western, and eastern regions of Africa led in terms of scientific publications during the period under review. The results further indicated that the continent experienced drought conditions in the years 1984, 1989, 1992, and 1997, thereby inducing an increase in the number of scientific publications on drought studies. The results show that the tools of analysis have also changed from simple statistics to the use of geospatial tools such as Remote Sensing (RS) and Geographical Information System (GIS) models, and recently Machine Learning (ML). The ML, particularly, contributed about 11% of the total scientific publications, while RS and GIS models, and basic statistical analysis account for about 44%, 20%, and 25% respectively. The integration of spatial technologies and ML are pivotal to the development of robust drought monitoring and drought prediction systems, especially in Africa, which is considered as a drought-prone continent. The research gaps presented in this study can help prospective researchers to respond to the continental and regional drought research needs.

**Keywords:** drought; monitoring; prediction; remote sensing; GIS; machine learning

## 1. Introduction

Drought is a naturally recurring phenomenon best characterized by multiple climatological and hydrological parameters. Due to increased water demand, exacerbated by factors such as rapid population growth, agricultural development, industrial and energy sectors development, water supply contamination, and climate change, water scarcity has continued to be a contending issue in many parts of the world [1]. The problem of water scarcity is further compounded by the occurrence of prolonged drought, affecting both groundwater and surface water resources. This is because drought often leads to the reduction in water supply, water quality deterioration, disturbance of riparian habitats, and crop failure. Drought is a water-related natural disaster which occurs when there are

continuous deficiencies in water supply, whether groundwater, atmospheric, or surface water. In [2], drought is defined as a slow and creeping recurring natural event, with its impact felt in numerous economic and social sectors. For instance, in the agricultural sector, drought is considered as one of the main causes of crop yield failure, particularly in both irrigated and rain-fed agro-ecosystems [3,4]. Similarly, an extreme drought event leads to the death of plants, animals, and humans. As suggested by [3], drought may be declared after 15 days of the continued shortage of either atmospheric, surface or underground water supply and such conditions can be prolonged for months or even years.

From the viewpoint of the frequency, duration, intensity, and severity, drought can be categorized as, meteorological, agricultural, hydrological, and socioeconomic [5]. These drought categories are often interlinked: e.g., meteorological drought can propagate into agricultural drought, hydrological drought, and socioeconomic drought; also, the longer the duration of drought the more spatially extensive it becomes [6]. Consequently, the attentions of scientists like meteorologists, agricultural scientists, hydrologists, and ecologists have been drawn to the topic of understanding drought characteristics through monitoring and predictions [7]. Therefore, studies on drought monitoring and prediction are essential for effective water resources management and planning.

Drought is a recurring, severe, and expensive event globally, due to climate change and extreme adverse natural occurrences [6], among other factors. For instance, since 1951, the duration, severity, and frequency of drought have increased, predominantly in Africa, southern Australia, and eastern Asia [8]. Global warming is expected to worsen the magnitude of drought at the regional level in the future [9]. According to [10], drought occurrence and effects vary from one climate regime to the other because it has a regional footprint. Drought is a frequently occurring phenomenon in Africa with a devastating impact on food security and consequently on human lives [11]. Based on the Emergency Events Database data (EM-DAT) [12] between 1900 and 2020, drought has affected over two billion people across the globe and has led to damages worth over 174 billion US dollars. About 44% of the total drought occurrences were recorded in Africa: this affected over 439 million people, caused the death of more than 867,000 people, and caused over 6.6 billion US dollars (see Figure 1) worth of damages. For instance, from 1900 to 2020, Ethiopia and Somalia experienced the highest drought occurrences in Africa. Other countries that experienced high drought episodes include Kenya, Niger, Mauritania, Mozambique, Burkina Faso, Chad, Mali, South Africa, Sudan, Cabo Verde, Senegal, and Tanzania. Nigeria is reported to have experienced fewer drought occurrences with about 3 million affected persons.

Drought indices quantify drought conditions through the integration of e.g., precipitation, rainfall, streamflow, and snowpack data into a logical big picture. Many drought indices have been developed over the years, some of the most commonly used are: Palmer Drought Severity Index (PDSI) [13], the Crop Moisture Index (CMI) [13], the Soil Moisture Drought Index (SMDI) [14], the Standardized Precipitation Index (SPI) [15], the Standardized Precipitation Evapotranspiration Index (SPEI) [16], the Effective Drought Index (EDI) [17], the Agricultural Reference Index for Drought (ARID) [18], soil wetness deficit index (SWDI; [19]), scaled drought condition index (SDCI; [20]); Microwave integrated drought index (MIDI; [21]), vegetation drought response index (VegDRI; [22]), and the Vegetation Health Index (VHI) [23]). These indices are usually formulated using sourced data from a numerical model, ground station, and remote-sensing data sources.

Detection, monitoring and mitigation of disasters entail the gathering of prompt and continuous applicable information that is effectively collected with field observations. Previously, drought events were monitored and predicted using observation data acquired from weather stations. Nonetheless, the uneven and sparse distribution of weather stations in developing countries increases the ambiguity in research that has to do with the evaluation of drought [24]. This problem has been solved using satellite remote sensing technology, from which large scale high-resolution climatic data can be easily acquired [25,26]. Satellite platforms enable the acquisition of frequent, current, near real-time, high-resolution spatial data [27]. In this regard, innovations in the fields of remote sensing (RS) and GIS have profoundly aided drought risk assessment over the last three decades. Most drought risk

assessment data changes over time, have spatial components, and are multi-dimensional. Thus, the use of RS and GIS has turned out to be indispensable.

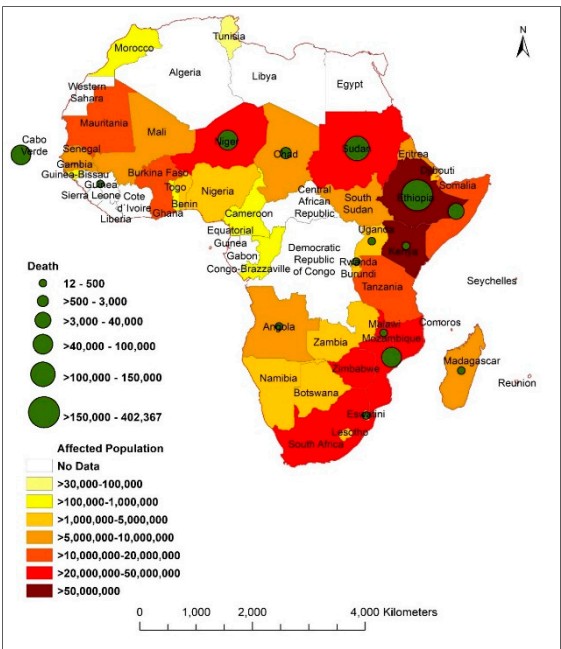

**Figure 1.** Droughts disasters by countries in Africa from 1900 to 2020 (Source: EM-DAT, 2020 http://www.emdat.be/database (access on 9 March 2020)).

Machine Learning (ML) is an application of Artificial Intelligence (AI) that offers systems the ability to automatically learn and improve from experience without being programmed. This methodology encompasses the science of statistical models and algorithms used by computer systems to progressively improve performance for reliable outcome [28]. The learning process begins with data or observations, like examples, direct experience, or instruction, to look for patterns in data and make better decisions in the future based on the examples that were provided. The principal aim is to allow the computers to learn automatically without human interference or help and regulate actions accordingly. Examples include Support vector regression (SVR) [29], least squares support vector regression (LS-SVR) [30], artificial neural network (ANN) [31], Deep belief networks (DBN) [32], Bagging [33], Random forest regression (RFR) [34], Adaptive neuro-fuzzy inference system or adaptive network-based fuzzy inference system (ANFIS) [35], Boosting [36], Hybrid models [37], Wavelet transform (WT) [38] and self-organising maps (SOM) [39]. A ML model is an output generated from data trained ML algorithm. Owing to the inherent architecture and the operating principle, the ML methods often use data to build a functional model or attribute that is a characteristic of the input data. The ML algorithms thus far have found wide scientific application spanning in such fields as earth system sciences [40], finance [41], and medicine [42]. They are also used to develop a comprehensive drought monitoring or prediction model [37].

Bibliographic methods can be defined as the statistical and mathematical approach of (quantitative) analysing (scientific) publications. It maps the history of publication, the physiognomies, and the progress of scientific output within a particular field of research [43]. Bibliographic techniques are suitable for categorizing and quantifying collaboration patterns of journals, publications, authors, institutes, and countries and can also be used to evaluate their contribution on definite topics [44]. The bibliometric techniques could be pragmatic at stages of titles, keywords list, publications summaries, or even the whole citation record to retrieve the specific topics and subject categories assigned to publications [45]. Not only does the co-occurrence of keywords suggest the assortment of research themes, but it also classifies the multidisciplinary character and directions (sub-areas/areas) for

additional advancement of a research field [46]. Bibliometric techniques are capable of drawing out vividly and graphically the leading topics, latest advances, and existing gaps in a certain field of a research discipline [46]. In addition, it plays an important role in the decision-making process related to science and it has been used by various authors [47].

Over the years, studies on drought have witnessed a rapid increase (manifested in the explosion scientific publications) due to the already increased frequency of the occurrence and associated impacts of droughts. This study is aimed at performing a bibliometric analysis of drought monitoring and prediction studies in Africa from 1980–2020. Given the aim, the following objectives were investigated; the relationship among drought occurrence and the progression of publication, critical review of methodology, and findings for the identification of research gaps over the African continent. As climate change potentially poses a great impact on the natural systems, information on drought and progress in available knowledge through research is crucial. This will help to improve understanding of the characteristics of occurrence, its impacts, and the development of strategic plans on how to manage and adapt to this natural disaster. Furthermore, the understanding of the existing body of knowledge on drought will pave the way for future research.

## 2. Materials and Methods

### 2.1. Materials

In this study, scientific documents on drought monitoring and prediction were retrieved from the two most commonly used core collection databases, the Web of Science (WoS) and Scopus. Both the WoS and Scopus are considered as the largest abstract and citation databases of peer-reviewed scientific articles, books, and conference proceedings covering a wide range of scientific disciplines including science, technology, social sciences, medicine, and humanities, among others. Subsequently, the two resources have been extensively used in bibliometric review research [48]. The use of both WoS and Scopus databases in this review helped to limit the risk of missing to capture certain documents in the field search [49]. In searching the database for drought monitoring and prediction research in Africa, various search topics were used [50]. For instance, these keywords were entered in both the WoS and Scopus web portals: "drought monitoring" OR "drought prediction" AND "machine learning" AND "remote sensing" AND "GIS" AND "Africa." These keywords were used in combination, e.g., "drought monitoring" AND "machine learning" AND "Africa," to search all the articles that were published in Africa on drought monitoring and those that utilized machine learning tool. The period of documents search covered almost 4 decades, i.e., 1980–2020. In total, 332 documents were retrieved from the WoS and Scopus core collection databases. Table 1 gives a summarized catalogue of the retrieved documents, based on the document type, e.g., articles, book chapters, conference paper, editorials, etc.

**Table 1.** Document types considered in drought monitoring and prediction mapping.

| Document Type | Number of Documents |
|---|---|
| Articles | 263 |
| Conference and proceedings papers | 44 |
| Book chapter | 9 |
| Editorial material and data paper | 5 |
| Letter | 1 |
| Reviews | 9 |
| Short survey | 1 |
| Total | 332 |

*2.2. Methods*

In this study, scientific data analysis was carried out based on bibliometric mapping techniques, i.e., the application of quantitative approaches to visually represent scientific information by use of bibliographic data [43]. Analysis of drought information was carried out based on bibliometric R package [43] whereas the visualization of bibliometric network maps was based on the VOSviewer (i.e., Visualization of Similarities) software [46]. In this research, the bibliometric analysis focused on the overall intellectual structure of drought monitoring and prediction and the application of tools such as spatial analysis (remote sensing, GIS) as well as machine learning during the selected study period. This was achieved by conducting a set of analysis that included annual production of scientific publications infield, the most productive countries and their collaborations, keywords occurrence, and thematic progression of research.

To assess the relationship among drought occurrence, the progression of publication, the methodology used in drought studies and the mean of cumulative 6-month SPEI, hereafter SPEI-6, was analysed over Africa divided into six regions excluding the northern regions (Algeria, Egypt, Libya, Morocco, Tunisia, and Western Sahara). The exclusion is as a result of the limited information and publication on drought. The classification was adopted from [51] and was based on the similarity in rain-bringing mechanisms and patterns of rainfall received since 1960 over the regions. The six regions include; Southern Africa (Botswana, Eswatini, Lesotho, Namibia, and South Africa); South Central (Comoros, Madagascar, Malawi, Mayotte, Mozambique, Tanzania, Zambia, and Zimbabwe); Central West (Angola, Congo-Brazzaville, and the Democratic Republic of Congo); West and East Gulf of Guinea (Benin, Cameroon, Central African Republic, Ivory Coast, Equatorial Guinea, Gabon, Ghana, Guinea, Liberia, Nigeria, Sierra Leone, and Togo); Horn of Africa (Djibouti, Eritrea, Ethiopia, Kenya, Somalia, and Uganda) and Sahel Sudan (Burkina Faso, Cabo Verde, Chad, Gambia, Guinea-Bissau, Mali, Mauritania, Niger, Senegal, Sudan, and South Sudan). The SPEI-6 which has a good correlation with agricultural and meteorological drought [16] was downloaded from the Spanish National Research Council and processed using R.

## 3. Results

*3.1. Scientific Mapping of Drought Monitoring and Prediction Research*

3.1.1. Trends in the Scientific Publications of Drought Monitoring and Prediction Research

Figure 2 depicts the annual distribution of published articles on drought monitoring and prediction in Africa between 1980 and 2020. Based on the results, research work on drought monitoring and prediction in Africa began to establish itself as an area of research interest in 1997. Since then, research on drought monitoring and prediction in Africa has exponentially increased- this is demonstrated by an overall annual percentage growth rate of scientific publications of approximately 6%. This suggests that the frequent occurrences of drought and its inherent impacts in most African countries reverberated a fast-growing interest amongst the scientific community to understand the underlying processes of drought, so to derive monitoring and prediction early warning drought systems, to plan for and mitigate future impacts of drought.

The rapid increase in the number of scientific publications on drought could be associated with the drought occurrences in Africa within the 6 regions using SPEI-6 shown in Figure 3. The figure depicts the areal-averaged of cumulative 6-months SPEI time series, for each of the six demarked sub-regions' presented in green (both positive and negative values) superimposed onto the continent's SPEI-6 presented in blue (wet periods-positive SPEI values) and in red (dry periods-negative SPEI values) indicates that the entire continent experienced drought conditions of the 80s and 90s in the years 1984, 1989, 1992, and 1997. Historically, the drought of 1973 (not shown) had a major impact in almost all African countries, north and south alike. Also, drought has persisted over the years in Africa and across the 6 regions since the year 1998 reaching "a Severe Dry" category of SPEI (−1.5 to

−1.99) in Western Africa (Figure 3B), with some intermittent occurrences of wet years except for the Sahel Sudan region (Figure 3D).

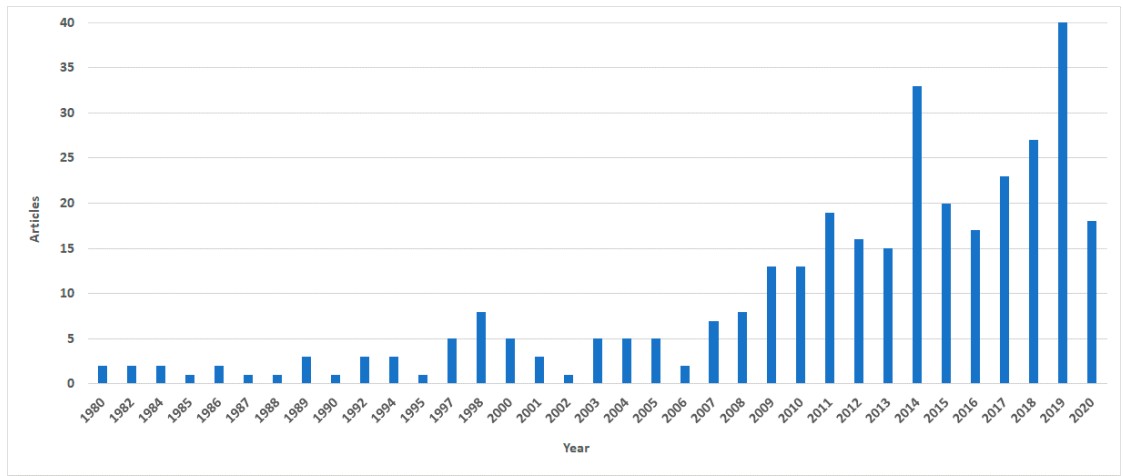

**Figure 2.** Annual distribution of publications in drought monitoring and prediction in Africa.

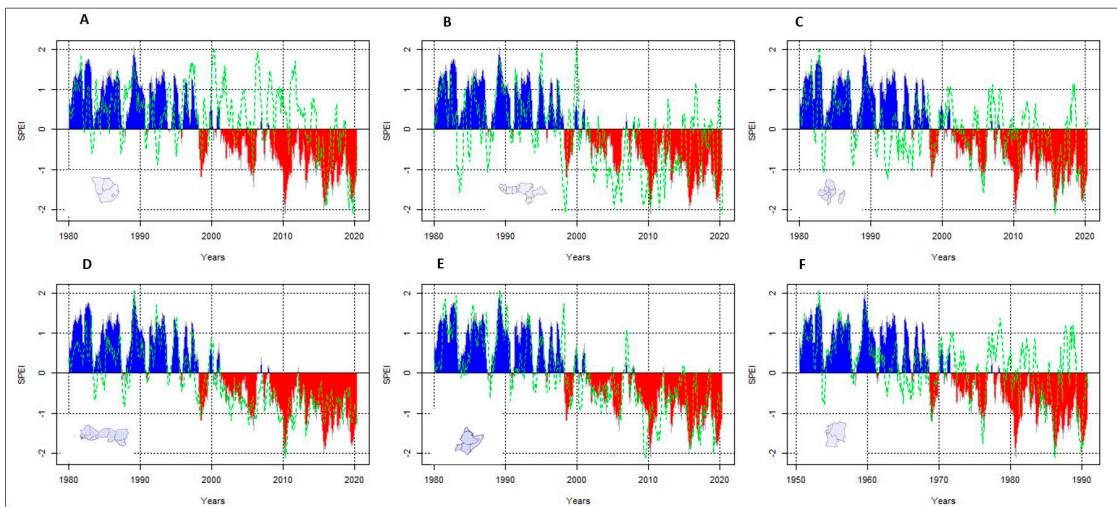

**Figure 3.** Timeseries of cumulative 6-month SPEI (SPEI-6) in Africa (filled blue and red) overlay with SPEI-6 in (**A**) Southern Africa (Botswana, Eswatini, Lesotho, Namibia, and South Africa); (**B**) West and East Gulf of Guinea (Benin, Cameroon, Central African Republic, Ivory Coast, Equatorial Guinea, Gabon, Ghana, Guinea, Liberia, Nigeria, Sierra Leone, and Togo); (**C**) South Central (Comoros, Madagascar, Malawi, Mayotte, Mozambique, Tanzania, Zambia, and Zimbabwe); (**D**) Sahel Sudan (Burkina Faso, Cabo Verde, Chad, Gambia, Guinea-Bissau, Mali, Mauritania, Niger, Senegal, Sudan, and South Sudan); (**E**) Horn of Africa (Djibouti, Eritrea, Ethiopia, Kenya, Somalia, and Uganda) and (**F**) Central West (Angola, Congo-Brazzaville, and the Democratic Republic of Congo).

The rapid increase in scientific publications on drought from 2007 as shown in Figure 2 is synonymous to continued drought occurrences in the continent illustrated in Figure 3. As shown in Table 2, the Southern Africa region comprising of Botswana, Eswatini, Lesotho, Namibia, and South Africa experienced 4 drought epochs each in the period 1980–2000 and 2001–2020 with varying maximum duration of drought (months) of 19 and 26 respectively. The region has the highest scientific publication on drought with a total of 83 having 10 (12%) from 1980–2000 and 73 (88%) from 2001–2020. West and East Gulf of Guinea (Benin, Cameroon, Central African Republic, Ivory Coast, Equatorial Guinea, Gabon, Ghana, Guinea, Liberia, Nigeria, Sierra Leone, and Togo) experienced 5 and 4 drought epochs, 23 and 51 months of drought duration with 7 and 41 scientific publications in 1980–2000 and

2001–2020 respectively. In the South Central (Comoros, Madagascar, Malawi, Mayotte, Mozambique, Tanzania, Zambia, and Zimbabwe) region, there were 5 and 4 drought epochs, 20 and 45 months of drought duration with 1 and 9 scientific publications in 1980–2000 and 2001–2020 respectively.

**Table 2.** Relationship drought occurrence and number of publications in Africa and 6 regions (1980–2020); following methods in [51].

| Region | 1980–2000 | | | 2001–2020 | | |
|---|---|---|---|---|---|---|
| | No of Drought Epochs | Max Duration of Drought (Months) | Total Number of Publications | No of Drought Epochs | Max Duration of Drought (Months) | Total Number of Publications |
| Southern Africa (Botswana, Eswatini, Lesotho, Namibia, and South Africa) | 4 | 19 | 10 | 4 | 26 | 73 |
| West and East Gulf of Guinea (Benin, Cameroon, Central African Republic, Ivory Coast, Equatorial Guinea, Gabon, Ghana, Guinea, Liberia, Nigeria, Sierra Leone, and Togo) | 5 | 23 | 7 | 4 | 51 | 41 |
| South Central (Comoros, Madagascar, Malawi, Mayotte, Mozambique, Tanzania, Zambia, and Zimbabwe) | 5 | 20 | 1 | 4 | 45 | 9 |
| Sahel Sudan (Burkina Faso, Cabo Verde, Chad, Gambia, Guinea-Bissau, Mali, Mauritania, Niger, Senegal, Sudan, and South Sudan) | 6 | 23 | 3 | 7 | '53 | 18 |
| Horn of Africa (Djibouti, Eritrea, Ethiopia, Kenya, Somalia, and Uganda) | 5 | 22 | 1 | 5 | 36 | 57 |
| Central West (Angola, Congo-Brazzaville, and the Democratic Republic of Congo) | 4 | 21 | 2 | 5 | 44 | 4 |
| Africa (Publications with the continent as the study area) | | | 22 | | | 84 |
| Total | | | 46 | | | 286 |

The Sahel Sudan (Burkina Faso, Cabo Verde, Chad, Gambia, Guinea-Bissau, Mali, Mauritania, Niger, Senegal, Sudan, and South Sudan) region recorded the highest drought epochs of 6 and 7, drought duration maxima of 23 and 53 months with 3 and 18 scientific publications in the years 1980–2000 and 2001–2020, respectively. In the Horn of Africa (Djibouti, Eritrea, Ethiopia, Kenya, Somalia, and Uganda) region, 5 drought epochs were recorded in each of the periods, having 22 and 36 months of maximum drought duration and 1 and 57 scientific publications in 1980–2000 and 2001 and 2020, respectively. Central West (Angola, Congo-Brazzaville, and the Democratic Republic of Congo) experienced 4 drought epochs in 1980–2000 and 5 epochs are 2001–2020. The region recorded 21 and 44 months of maximum drought duration with 2 and 4 scientific publications for the periods of 1980–2000 and 2001–2020 respectively. In summary, there is an increase in the numbers of drought epochs, durations and scientific publications across the regions from the 1980–2000 to 2001–2020 period. The Sahel Sudan region has the highest drought epochs (12) and maximum duration of drought (76 months) but had only 21 scientific publications compared with the Southern and Western regions which had lesser numbers of drought epochs and durations but with more scientific publications. Research publications with the African continent as the study area accounts for about 46% of the total publications.

As stated in the materials section, 332 documents on drought monitoring and prediction were retrieved from WoS and Scopus datum and analysed in the current study. Several countries contributed to the publication of these articles. Table 3 gives a summary of the 15 top countries that have contributed to research work on drought monitoring and prediction in Africa. As it can be seen from the table, South Africa is the most publishing country with 56 published articles, 45 of those published under single country inter-collaborations and 11 published through multiple country collaborations. The USA, takes the second lead with 54 articles comprising of 46 single country publications and 8 multiple country publications. It is worthwhile to mention that the ranking of countries depends on the affiliation of the main author in the article.

**Table 3.** Most productive countries in the production of drought monitoring and prediction articles.

| Country | Articles | Single Country Publications | Multiple Country Publications |
|---|---|---|---|
| South Africa | 56 | 45 | 11 |
| USA | 54 | 46 | 8 |
| Germany | 16 | 14 | 2 |
| United Kingdom | 12 | 6 | 6 |
| France | 11 | 9 | 2 |
| Italy | 10 | 10 | 0 |
| Canada | 8 | 7 | 1 |
| Ethiopia | 8 | 6 | 2 |
| Kenya | 8 | 6 | 2 |
| Nigeria | 8 | 4 | 4 |
| Zimbabwe | 8 | 3 | 5 |
| China | 7 | 4 | 3 |
| Netherlands | 7 | 5 | 2 |
| Belgium | 7 | 3 | 2 |
| Australia | 5 | 1 | 4 |

### 3.1.2. Country Collaborative Analysis

The VOSviewer program software was used to map and visualize bibliometric collaboration network between countries that have published scientific documents on drought monitoring and prediction over the period of study. In the analysis, the program assigned 63 countries into clusters. Results of bibliometric collaboration network of countries are depicted in Figure 4. In this analysis, the countries were assigned to 11 clusters, each of the country assigned once. The countries are indicated by the colour of the cluster to which they belong. Countries' collaboration is described by the lines (links) that joins the countries. In this analysis, only 300 links representing the 300 strongest links between countries are displayed, as most are either embedded within the strong ones or too faded to be seen. In addition, the greater the size of the cluster the stronger is the collaboration between the countries. For instance, South Africa in the green cluster shows the strongest collaboration with other countries, e.g., it exhibits 56 links, followed by the USA and Nigeria with 55 and 49 links, respectively. The leading collaborative countries in the red cluster are the Netherlands (51 links), China (51), France (50), and Australia (46). The UK, Spain, and Belgium are the main collaborative countries in the blue cluster. Significant collaborations between countries are shown under the yellow cluster, with Germany (54 links), Ethiopia (49), Kenya (47), and Zimbabwe (46) taking the lead. Furthermore, Singapore, Brazil, and Malaysia are the leading collaborative countries under the purple cluster with 37, 35, and 33 links, respectively.

### 3.1.3. Keywords Analysis

Figure 5 depicts results on keywords co-occurrence analysis. These are keywords extracted from the titles or abstracts of the 332 analysed data. Similar, to country collaboration analysis in VOSviewer, the keywords were grouped into 7 clusters and analysed based on the strength association methodology. The clusters are represented by circles of different colours and size. Based on the results, four clusters are of significant size, suggesting a strong co-occurrence of the corresponding keywords in either the titles or abstracts. These clusters include a yellow cluster with the following keywords: drought, soil moisture, and monitoring; the green cluster containing the keywords such as climatology, drought monitoring, precipitation/rainfall; the red cluster covering keywords like agriculture, climate change,

environmental monitoring, geographical information, and GIS; the blue cluster with keywords such as water management and arid/semi-arid region; and the purple cluster with Africa, precipitation intensity, crop adaptation, prediction, climate, and environmental impact assessment.

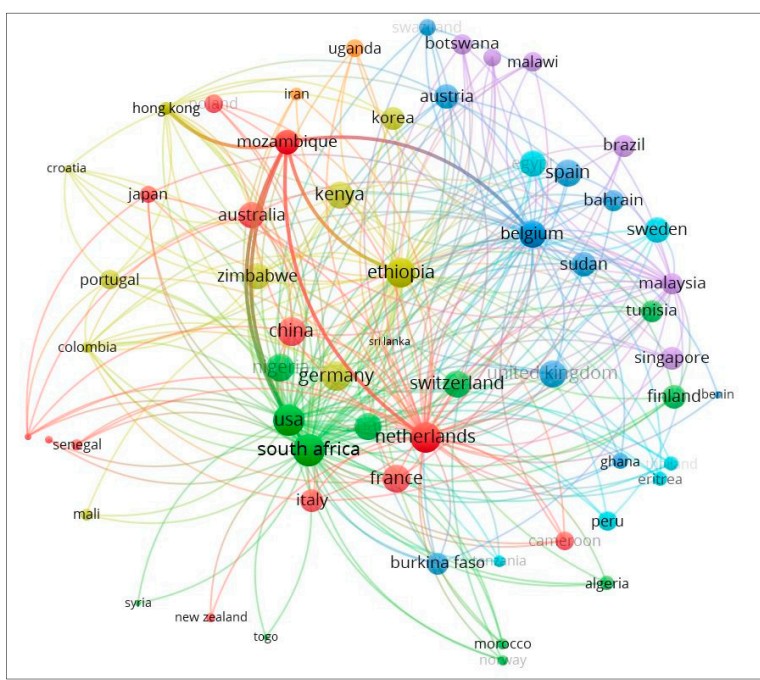

**Figure 4.** Bibliometric graphical analysis of countries collaboration network of drought publications in Africa (1980–2020).

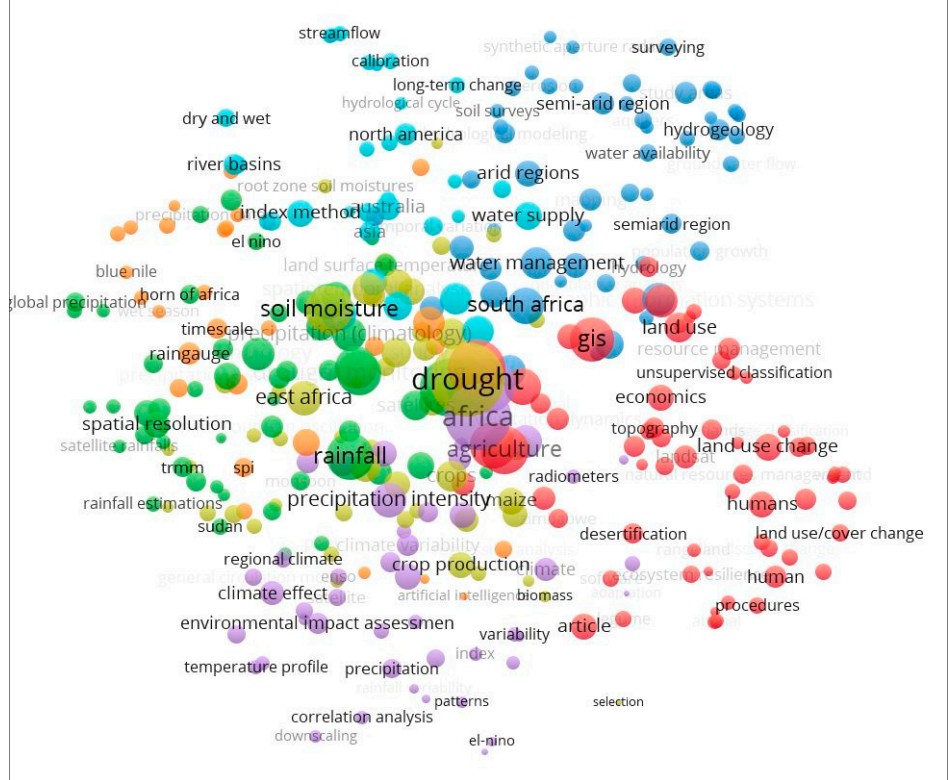

**Figure 5.** Bibliometric graphical analysis of keywords co-occurrence analysis of drought publications in Africa (1980–2020).

### 3.1.4. Thematic Progression Analysis

Figure 6 illustrates the key research themes based on scientific publications on the subject of drought monitoring and prediction in Africa. Four themes quadrant is presented going by their centrality and density rank values along with horizontal-axis centrality and vertical-axis density [52]. The dimensions of the spheres (circles) are proportional to the number of documents/publications equivalent to each keyword in each quadrant. Themes located in the upper-right quadrant are called the motor-themes of the studies (represent the hot topics), as they present high density and strong centrality. The location of themes in this quadrant indicates that they are related visibly to concepts pertinent to other themes that are conceptually closely related. Themes located in the lower-right quadrant are called the general and transversal, basic themes. The quadrant specifies keywords that are vital in the research field (drought monitoring and prediction) but are not developed. Additionally, the themes located in the lower-left quadrant are called either disappearing or emerging themes. The quadrant specifies research studies that are weakly developed and marginal with low centrality and low density. Themes in the upper-left quadrant are very focused and peripheral. They possess well developed internal ties but insignificant external ties and thus they are of only marginal relevance to the field. Consequently, the scientific publications given in Figure 5 point to drought studies that focus on water management and decision making present high density and strong centrality indicating that they are related visibly to other themes that are conceptually closely related. As displayed in the lower-right quadrant, drought research utilizing remote sensing and GIS techniques and assessment of the impact of rainfall variability, climate change, and trends for drought monitoring is the basic vital research fields, but are not yet fully explored. The emerging or disappearing themes located in the lower-left quadrant comprises drought studies of dry spells and the use of models. The themes located in upper-left quadrant illustrate drought studies that are of marginal importance considered in Africa over the period under review. The quadrant shows drought studies focusing on the use of spatial analysis, climate change vulnerability; the use of evapotranspiration dataset; impact analysis on maize crop, biomass production; forecasting and international cooperation.

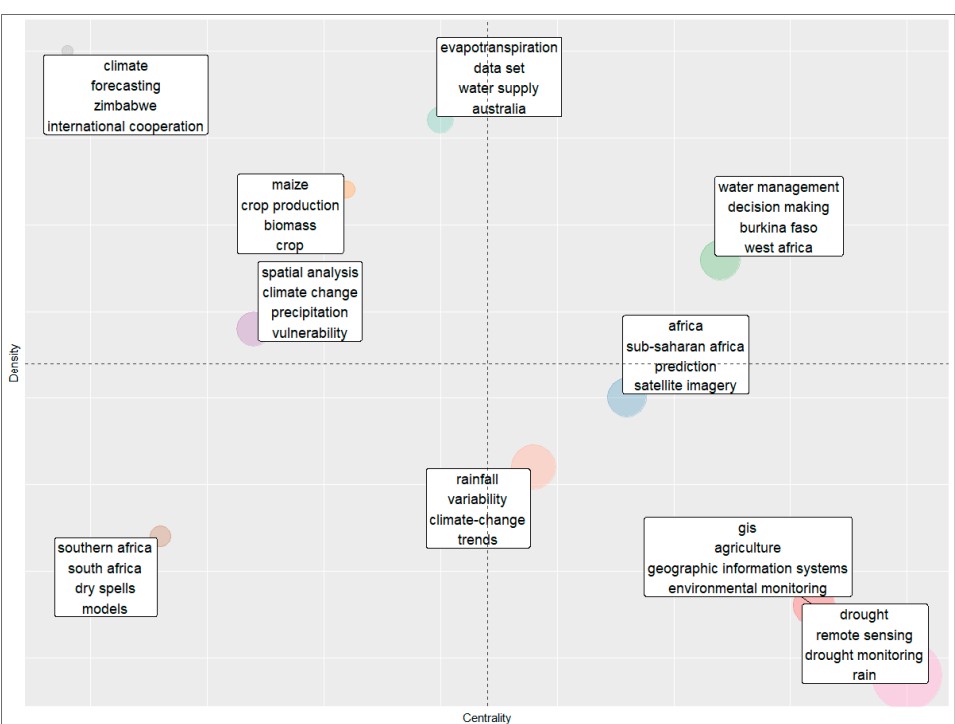

**Figure 6.** Thematic progression of drought research in Africa (1980–2020).

### 3.1.5. Direct Citation Analysis

To illustrate the progression of drought studies, methodologies used and identify research gaps, only journals that have direct connectivity or citation with each other were reviewed. A summary of the drought and prediction study are displayed in Table A1 (provided in Appendix A). Figure 7 illustrates the historical direct citation network for drought monitoring and prediction studies in Africa from 1980–2020. The clustering demonstrates an assemblage of articles having a direct citation. In this regard, ten clusters were identified. A comprehensive review of the historical direct citation network (Figure 7) was conducted in order to tract the evolution of drought monitoring and prediction knowledge. The first 2 earliest scientific publications and 3 latest scientific publications in the cluster were referenced (Table A1 in Appendix A). However, for some clusters, 1 earliest and 2 latest are also referenced.

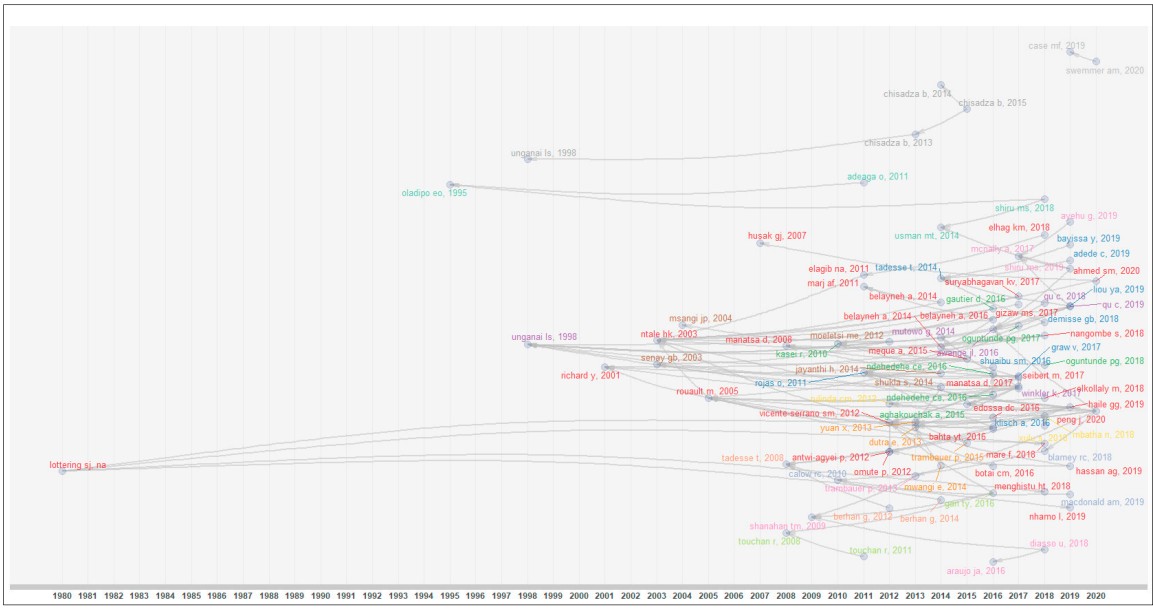

**Figure 7.** Historical direct citation network of drought publication in Africa (1980–2020).

As depicted in Figure 7, studies and therefore scientific publications on drought by, for instance, the work of [53] emerged in 1980 (red coloured cluster). The study analysed the impact of drought on small scale farmers using a combination of quantitative and qualitative research methods. The study was directly cited in 4 other studies (Figure 7; 4 direct lines) and thereafter by [54–57]. In general, the studies were centrally focused on quantifying the impact of drought on agriculture and food security. Studies in this cluster used several methods that include the use of structured questionnaires in 1980 [53] to assess the impact of drought on small scale farmers, the assessment of geospatial tools to monitor drought in 2012 [54], geospatial analysis of SC-PDSI in 2016 [55], Normalized Difference Vegetation Index (NDVI) [56], and SPEI in 2020 [57]. The studies reported varying findings that include the positive relationship socio-economic and drought, the importance of indigenous knowledge for adaptive strategies, the advantages offer using geospatial tools and a high correlation between NDVI and drought. The use of drought information tools and the implementation of drought management plans to support real-time decision-making and studies on adaptive measures for drought were recommended for future research.

The second cluster is led by the work by [58] in 1995 (Lime colour cluster) and is cited by [59–61]. Studies by [58] focused on examining the interannual variability of drought areas. Generally, the studies are related to drought monitoring. The researchers employed various methods such as Bhalme-and-Mooley-type drought indices and statistical tests in 1995 to the SPI in 2011, the intra-seasonal rainfall monitoring index (IRMI) in 2014, and SPEI in 2018. The studies reported various

findings, among which is monitoring of the rainfall regime particularly during its onset phase [60] and they suggested more in-depth studies on extreme drought, interannual variability, and trend analysis for future drought occurrence.

The grey cluster was led by the work of [62] in 1998 and was cited by [63–65]. Studies by [62] investigated the application of Advanced Very High Resolution Radiometer (AVHRR) data in regional scale drought monitoring. In general, the studies focused on agricultural drought in southern Africa. The studies made use of various methods which include simple statistical analysis to analyse structured questionnaires [63], to the use of remote sensing (NDVI, vegetation condition index (VCI), and temperature condition index (TCI)) [62,65] to model comparison [63]. The findings of the studies include the integration of traditional drought forecasting with meteorological forecasting to guarantee sustainable rural livelihood development. Research gaps identified were the need to improve the accuracy of the drought monitoring and prediction systems and the understanding of the drivers of drought variability and trends.

The fourth cluster (lavender colour) is led by the research of [66] in 1998 and was cited by [67–70]. The studies by [66] focused on monitoring drought and corn yield estimation. Generally, the studies are related to seasonal variation of drought using remotely sensed data to derive several vegetative and drought indices. Correlation and trends analysis were the major analysis performed. Their findings illustrated the usefulness of remote sensing technology for effective drought monitoring. Some of the recommended and conceived research gaps for further studies were the need to investigate the severity of vegetation stress and extreme drought at a regional level and for future decades.

The brown colour cluster starts with the studies by [71] in 2003, cited by [72–74]. The study by [71] aimed at characterizing yield reduction. Studies in the cluster are centred on agricultural drought risk. The following methods were used for these studies GIS-based crop water balance model [71], to statistical analysis [72], to remote sensing [73], and hydrologic model [74]. Their findings include the development of seasonal agricultural drought forecasts to improve water and agropastoral management decisions, support the optimal allocation of water resources, and mitigate socioeconomic losses incurred by floods and droughts. Suggested future research includes agricultural drought risk profiling analyses for other crop types and the calculation of seasonal crop water balances.

As shown by the green colour cluster, the studies by [75] in 2010 were cited by [76–79]. The studies investigated the temporal characteristics of meteorological droughts in the Volta Basin. The studies located in this cluster are related to hydrological drought. In general, the studies in this cluster made use of a range of satellite derived drought indices. Many of these studies responded to some of the earlier research gaps in the application of geospatial data and analysis to drought studies. Future research on the efficiency and effects of drought responses on populations and interactions among the different responses were recommended.

The gold colour cluster is preceded by the studies reported by [80] in 2013 and cited by [81,82]. These studies evaluated the use of European Center for Medium-Range Weather Forecast (ECMWF) products in monitoring and forecasting drought conditions [80]. Studies in this cluster are mainly focused on drought prediction using ECMWF. Their researches output which is forecasting the spatial extent and intensity of the drought event seem to have responded to previous research gaps identified in earlier researches. Further studies on hydrological drought seasonal forecast and skill assessment, meteorological seasonal forecast, and global monitoring and forecasting of drought were proposed.

The yellow colour cluster studies emerge with the work of [83] in 2012 focused on characterizing agricultural drought. Generally, the studies in this cluster utilized satellite derived variables and mathematical modelling [83–85]. Many of the studies improved on earlier drought studies on the application of geospatial technologies. In particular, new vegetative indices such as the burned area index (BAI) and normalized difference infrared index (NDII) were developed and were reported to be more reliable for drought monitoring than the traditional NDVI. Research gaps identified include the proposal for further studies on vegetation response pattern to drought, policy and adaptation strategies development.

In the blue-grey colour cluster, first studies were led by [11] in 2011 and cited by [86,87]. The studies by [11] proposed a novel method for calculating the empirical probability of having a substantial proportion of the entire agricultural area affected by drought at the sub-national level. The studies in this cluster are related to agricultural drought and utilizing remote sensing. The studies in this cluster responded to previously identified gaps by exploring the use of vegetative indices to detect changes in total annual vegetation productivity and monitoring drought dynamics during the season using VCI. They suggested future drought monitoring in Africa could be based on drought occurrence over both the temporal and the spatial domain.

The indigo colour cluster has the work by [88] in 2014 as the first publication and cited by [89–91]. The research aimed at developing an experimental drought monitoring tool that predicts the vegetation condition (vegetation outlook). Studies in this cluster are centred on agricultural drought prediction. Their methods include regression-tree technique [88], classification and regression tree (CART) modelling technique [89], ML, and general additive model (GAM) technique [90], and Vegout-UBN model [91]. Among their findings is the ex-ante drought early warning systems capable of offering drought forecasts with enough lead times. The authors suggested improvement of both the administrative and spatial resolution of the model to determine drought status at district levels for effective drought monitoring and prediction.

In general, this review indicates that 75% of the studies were related to agricultural (55%) and hydrological (20%) drought. The remaining 30% is shared among socioeconomic, meteorological, and impact studies. The results further revealed that methodologies used in drought studies among researchers in the African continent have evolved. This growth is partly linked to the various responses to the gaps in drought studies. The review indicated that methodologies evolved from the use of simple statistical analysis to analyse historical climatic data and drought indices such as SPI in the 80s to late 90s, to the use of spatial tools such as remote sensing and GIS to analyse vegetative indices such as NDVI and other emerged drought indices during the era of the late 90s and early 2000 with the research of [66] in 1998. The review suggested that the use of models began in early 2000 with the studies by [71] in 2003. Besides, the results indicate that as drought persists in the continent with severing impacts, researchers are engaging more sophisticated methods that include the use of ML for drought prediction with the recent study of [92] in 2000. This review indicates that the application of ML approach in drought studies emerged in 2014 (as demonstrated by the 4 scientific publications) and a significant increase its application by 2019 (because of the 8 scientific publications). In all, ML contributed about 11% of the total scientific publications, while geospatial technologies (all spatial analysis including RS and GIS), models, and basic statistical analysis account for about 44%, 20% and 25% respectively. Worthy to mention that there was also a hybrid of methodological approach where some researchers have combined two or more drought analysis frameworks.

*3.2. Some Perspectives of Drought Monitoring and Prediction Intelligence Tools*

The scientific networks and mapping that underpin four decades of drought monitoring and prediction research presented in Section 3.1 point to two main perspectives i.e., (a) the advances in drought monitoring and prediction indicators from a climatological standpoint (this includes assessments of anomalies in variables such as precipitation, soil moisture, evaporation), and (b) and the analysis of drought impacts/risks from an ecosystem perspective (for example in the use of agro-hydrological anomalies). In both viewpoints, drought monitoring and prediction have benefited from the increasingly quantitative and qualitative spatio-temporal resolution data sets gathered from terrestrial and space-based technologies as well as Numerical Weather Prediction (NWP) models or global climate models.

The multitude of the observational networks as well as data from NWP model simulations gave rise to databases with archived data sets dully suited for the development of drought early systems. Coupled with the development of observational networks and improved NWP models simulations, intelligent drought monitoring and prediction systems have emerged, benefiting from the geospatial

technologies and ML. While these advanced methods and tools have widely been utilized in the development of drought early warning systems globally, the application of ML for drought research as remained modest in the African continent. This is evidenced in the 10.5% representation of the total drought monitoring and prediction scientific publications focusing on Africa in the last four decades (see Figure 8). Whereas the drought conditions continue to ravage the African continent, a subtle annual average production of two scientific publications, notwithstanding the gradual progression, during this period demonstrates that the use of intelligence tools (which utilize the ML technique) for drought monitoring and prediction in the continent is still at its infancy.

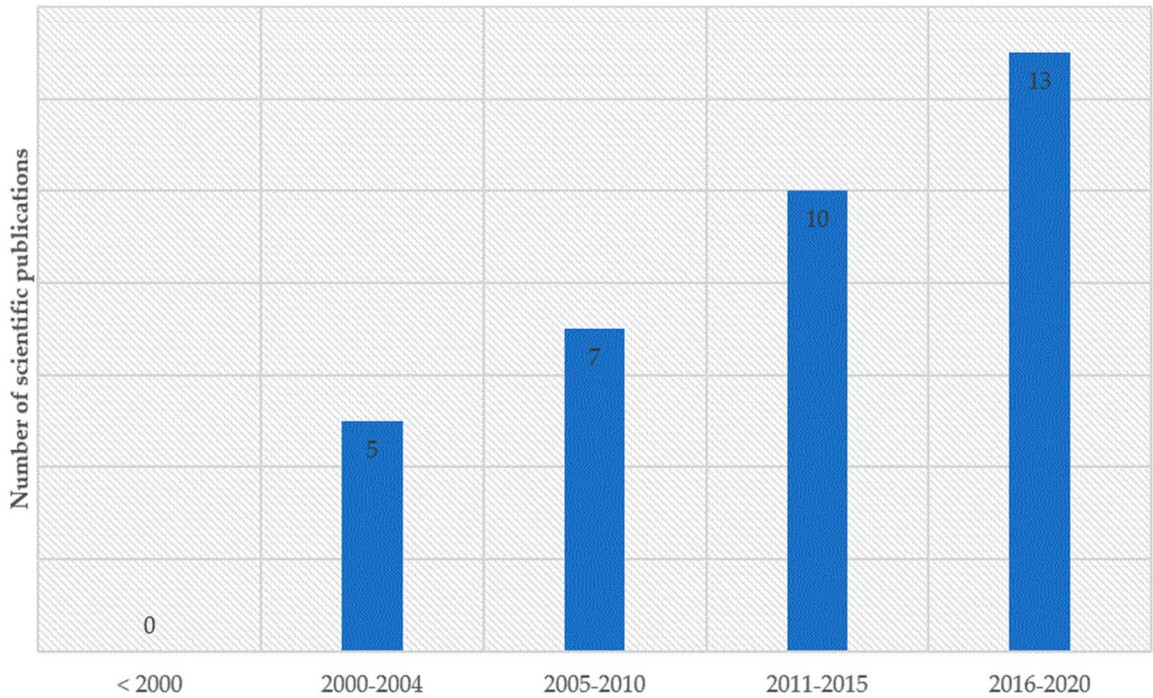

**Figure 8.** Growth in Machine Learning scientific publications.

## 4. Discussion

The data from EM-DAT illustrates the impact of drought occurrence in Africa, particularly in the Southern, Eastern (Horn), and west-central regions of the continent. This review revealed that the frequent occurrences of drought and its inherent impacts in most African countries led to a fast-growing interest amongst the scientific community to understand the underlying processes of drought. This is coupled with the quest to develop robust early warning drought systems that could be used to plan for and mitigate future impacts of drought. This result corresponds to the study of [6] which demonstrated the growing interest in the application of models to drought monitoring and prediction in the early 2000s.

According to this review, researchers have explored various aspects of drought (that is meteorological, hydrological, agricultural, and socioeconomic drought). As shown in this review, agricultural and hydrological drought studies contributed about 75% of the total scientific publications. The remaining 25% is shared among socioeconomic and meteorological studies. The researchers made use of various indices: the NDVI (31% of the reviewed studies) and SPI (23% of the reviewed studies) were frequently used. This can be attributed to the advent of vegetative indices in studying drought and simplicity of required input data for the calculation of the SPI [15].

As shown in the time series of the SPEI-6 and its comparison with drought research scientific publications, the results reflect a relationship in the progression of drought occurrence and scientific publications. This relationship is particularly important to establish the "chain reaction" of researchers

to drought occurrence. The Sahel Sudan region has the highest drought epochs (12) and duration (76) over the study period. This can be attributed to the climatic characteristics of the region. However, the research outputs do not seem to correspond to the drought occurrence when compared with the Southern and Western regions which had lesser numbers of drought epochs and durations but with more scientific publications. As shown in Figure 2, the first set of scientific publications on drought emerged in 1980 [43]; this could be attributed to the 1973 drought. Increased drought research publication in the 2000s can be seen in the light of the persistent drought from the year 2000. Furthermore, the dominance of drought research publications from the Southern Africa (83), Horn of Africa (58), and West of the Gulf of Guinea (48) can be attributed to the severity and persistence of drought in the regions. It could also be attributed to assess to improved research tools, international collaboration, and funding.

The progression of methods and tools used for drought studies in the continent can also be linked to the advancement of the spatio-temporal time series analysis. In order to better detect, monitor, and predict drought conditions, drought characterization transcended simple statistics to modelling with the use of geospatial tools such as RS and GIS, and recently the ML approach. In this regard, the use of statistical tools to analyse drought indicators were reported by studies in the 1980s. This could be due to dearth of the understanding of drought, required skills, and data availability. As drought persisted in the continent with severe impacts, more sophisticated methods that include the use of models and geospatial tools and ML to provide answers to research gaps. Although, geospatial tools have been widely used elsewhere and in various fields, this review indicates that the use of geospatial tools in drought studies in Africa emerged in 1998. This could be related to the fact that some of the indices used for drought studies are satellite-derived indices such as NDVI and offers a spatial dimension to the monitoring and prediction of drought. This result synchronizes with the findings of [7], who reported several concepts used in drought studies. The study indicated the wide use of drought indices, combined with remotely derived indices and models. It is important to note that the use of ML for drought monitoring and prediction only emerged in the early 2000s. This could be attributed to the enormous data sets, growing quest to develop an improved drought predicting system in the eye of the persistent drought further exacerbated by the climate change. The ML method has an advantage for resolving complex phenomenon such as the drought that requires multiple inputs. Studies on the use of ML have reported that ML has a higher capability of high accuracy rate for prediction [90].

The identified research gaps were usually related to the conducted research topic, which implies that it will be addressed under similar themes. But, the gaps could also spark the innovation of other research areas of drought. Some of the identified research gaps were addressed by the researchers themselves in their subsequent studies like in the case of [63] who suggested the integration of traditional drought forecasting with meteorological forecasting to guarantee sustainable rural livelihood development and they address this in [64,65]. Other identified research gaps were addressed by other researchers in another drought research area; for instance, the recommendation for future drought monitoring in Africa could be based on drought occurrence over both the temporal and the spatial domain by [11] was addressed by [68]. Some of the identified research gaps have been addressed while some are outstanding, especially the recent studies. The identified research gaps provide the state of the art of knowledge and inform the future course of study. Hence, they can help prospective researches to readily respond to the various research needs. In general, this review further exposes the magnitude of drought impact in Africa and echoed the findings of [56]. This review further provides an overview of the state of research response, where future research should be focused and what appropriate tools to engage.

## 5. Conclusions

Drought in Africa has been investigated, predominantly, from meteorological, agricultural, and food security perspectives. Nonetheless, the studies do not cover all the African countries that are prone to drought incidences. Because of the enormous devastating effects of droughts on the continent,

it is opined that there has been an upsurge in drought studies focusing on the African continent. This review concludes that drought studies have evolved from using field observation to using the advanced technology technique (RS and GIS) for data gathering, sorting, manipulating, and analysing for better understanding of the subject matter.

The use of spatial tools such as remote sensing and GIS for drought studies is averred as a fundamental research methodology not yet fully explored. There is a need to focus more on the use of these tools given the advantages offered from them. The present review determines that the use of models and machine learning for drought monitoring and prediction are emerging themes. The advent of ML coupled with spatial models has potential to provide the adequate tools needed for the development of an integrated drought early warning system for the continent and could allow for easy scaling to regional and local levels. The scalability and high accuracy of monitoring and prediction of drought are identified as the major future research gap to be filled.

The marginal studies on international collaborations points to the in-exhaustive international cooperation on drought studies or management in the African continent, a vital research network needed now more than ever. This can be viewed as a major limitation to harnessing resources that exist internationally and reminiscent among developed nations. It is important to note that the review highlighted some drought studies that were premised on international collaboration (outside Africa) that were funded by international donors or funders such as the World Meteorological Organization (WMO), the Food and Agriculture Organization (FAO), and the United States Agency for International Development (USAID). Research studies needed to develop tools such as an early warning system are very vital for drought monitoring and prediction but are costly. Hence, cooperation between and among African nations coupled with international funding to support the data-intensive and high skilled requirement for studies such as drought are needed.

**Author Contributions:** This study was conceptualized by O.M.A. The methodology was developed by O.M.A., M.M., J.O.B and CMB. Formal analysis were performed by O.M.A., J.O.B., and C.M.B. The original draft was written by O.M.A. and M.M., J.O.B. and C.M.B. edited the draft. All authors have read and agreed to the published version of the manuscript.

**Funding:** The project reported in this paper is partially funded by SIDA, through the Swedish International Development Cooperation Agency, the Foreign Ministry of the Netherlands, the South African Department of Science and Technology, and USAID under Award No. AID-OAA-F-17-00034 Under Securing Water for Food: A Grand Challenge for Development FRA Number: SOL-OAA-16-000176. The project is also partially funded by the South Africa Research Foundation (NRF) grant for 2019: Thuthuka Funding Instrument (Unique Grant No: 117800).

**Conflicts of Interest:** The authors declare no conflict of interest.

## Appendix A

**Table A1.** A summary of selected scientific publications with direct citation of drought monitoring and prediction studies conducted in Africa 1980–2020.

| Studies | Location | Aim | Methods | Research Gaps | Type | Cluster Color, See Figure 7 |
|---|---|---|---|---|---|---|
| [53] | South Africa | To assess the impacts of drought and the adaptive strategies of small-scale farmers | questionnaires, focus groups, and key informant interviews | Studies on adaptive measures for drought | DM | Red |
| [54] | Africa | To illustrate how the development of drought information systems based on geospatial technology could improve the possibilities of drought mitigation in Africa | drought information systems based on geospatial technology | The use of drought information tools fundamentally the implementation of drought management plans and to support real-time decision-making. | DM | Red |
| [55] | South Africa | to identify and characterise drought events | self-calibrated palmer's drought severity index (SC-PDSI). | Further studies on identification and characterization of drought event | DM | Red |
| [56] | East Africa | an overview of drought studies | Existing studies | Output from the study will form a basis of information for other regions outside of East Africa | DM | Red |
| [57] | Africa | to assess drought vulnerability considering a multi- and cross-sectional perspective | normalized difference vegetation index (NDVI), SPEI | Use of SPEI-HR for the study of drought-related processes and societal impacts at sub-basin and district scales in Africa. | DM | Red |

**Table A1.** *Cont.*

| Studies | Location | Aim | Methods | Research Gaps | Type | Cluster Color, See Figure 7 |
|---|---|---|---|---|---|---|
| [58] | Nigeria | To examine the interannual variability of seasonal Bhalme-and-Mooley-type drought indices | Bhalme-and-Mooley-type drought indices, Statistical tests | Further studies on interannual variability of the drought | DM | Lime |
| [59] | Nigeria | To investigate drought episodes | SPI | More in-depth studies on extreme drought | DM | Lime |
| [60] | Nigeria | To propose a simple drought monitoring and early warning (ew) methodology | intra-seasonal rainfall monitoring index (IRMI) | Stern monitoring of the rainfall regime particularly during its onset phase | DM | Lime |
| [61] | Nigeria | To assess the impacts of recent climate changes on drought-affected areas and drought incidence during different cropping seasons | standardized precipitation evapotranspiration index (SPEI), Statistical analysis | Trend analysis for future drought occurrence | DM | Lime |
| [62] | Southern Africa | To investigate the possible application of AVHRR data in regional scale drought monitoring | NDVI, vegetation condition index (VCI), and temperature condition index (TCI) | The areal extent and core areas of recent droughts can be further mapped and validated by mean atmospheric anomaly fields | DM | Grey |
| [63] | Zimbabwe | To detect, evaluate, and document local traditional indicators used in drought forecasting in the mzingwane catchment and to assess the option of integrating traditional rainfall forecasting, using the local traditional indicators, with meteorological forecasting methods | structured questionnaires, statistical analysis | integration of traditional drought forecasting with meteorological forecasting to guarantee sustainable rural livelihood development. | DP | Grey |
| [64] | Zimbabwe | To verify the applied local traditional knowledge (LTK) indicators in Mzingwane catchment and validate their accuracy and reliability in drought forecasting and early warning | structured questionnaires administered, hind-cast comparison | more validation should be carried out for several seasons, in order to standardize the LTK indicators per geographical area | DP | Grey |
| [65] | Southern Africa | To assess linkages between selected local traditional knowledge (LTK) indicators with meteorological drought forecasting parameters. | structured questionnaires administered, standard precipitation index (SPI), trends, normalized difference vegetation index (NDVI) | constant monitoring and standardization of LTK data | DP | Grey |
| [66] | Southern Africa | To detect, map and track the temporal and spatial characteristics of the drought, and estimating usable corn yield | vegetation condition index (VCI) and temperature condition index (TCI) | VCI and TCI can be used for further studies on detection, tracking and mapping of temporal and spatial characteristics of drought | DM | Lavender |
| [67] | Zimbabwe | To analyze the spatial variations in the seasonal occurrences of drought | NDVI, VCI | remote sensing technologies employing indices such as the VCI is competent for drought monitoring | DM | Lavender |
| [68] | Horn of Africa | To examine the application of spatial independent component analysis (SICA) and extract distinct regions with similar rainfall and total water storage (TWS) | SICA, SPI, total storage deficit indices (TSDI), standard precipitation indices (SPIS), Correlations analysis | Meteorological drought impacts can be based on TWS changes resulting in several years of extreme hydrological droughts. | DM | Lavender |
| [69] | Horn of Africa | To investigate the impacts of extreme agriculture drought and food security | NDVI, VCI, TCI, VHI and trends | Further studies can be based on discovering the spatial patterns and temporal trends of vegetation stress and extreme drought events at regional level. | DM | Lavender |
| [70] | Horn of Africa | To examine extreme drought | NDVI, VCI, TCI, vegetation health index (VHI), trends and correlation analysis | Further studies can be carried out to demonstrate the severity of vegetation stress and extreme drought for future decades | DM | Lavender |
| [71] | Ethiopia | To characterize yield reduction | GIS-based crop water balance model | using geospatial rainfall estimates derived from satellite and gauge observations, where available, seasonal crop water balances can be developed | DM | Brown |
| [72] | Southern Africa | To examine suitable drought mitigating initiatives, relating them to land tenure and land management practices. | Existing studies | Expansion of this type of study at a global scale. An informed global action is required. | DM | Brown |
| [73] | Sahel | To highlight the consequences of agricultural drought risk profiling analyses for maize | water requirement satisfaction index (WRSI) | Agricultural drought risk profiling analyses for other crop types | DM | Brown |
| [74] | East Africa | To describe the development and execution of a seasonal agricultural drought forecast system | variable infiltration capacity (VIC) hydrologic model, WRSI, statistical analysis, ESP | More accurate seasonal agricultural drought forecasts for this region can help update improved water and agropastoral management decisions, support optimal allocation of the region's water resources, and mitigate socioeconomic losses incurred by floods and droughts. | DP | Brown |

**Table A1.** *Cont.*

| Studies | Location | Aim | Methods | Research Gaps | Type | Cluster Color, See Figure 7 |
|---|---|---|---|---|---|---|
| [75] | West Africa | To investigate the temporal characteristics of meteorological droughts in the Volta basin | Standardized Precipitation Index (SPI) | More research needed on extreme drought conditions | DM | Green |
| [76] | East Africa | To outline a framework for using ensemble streamflow prediction (ESP) concept for multivariate, multi-index drought prediction | SPI, multivariate standardized drought index (MSDI), ensemble streamflow prediction (ESP) | Application of satellite precipitation data for regional to global drought monitoring | DP | Green |
| [77] | West Africa | To assess hydrological drought characteristics over the basin | SPI, standardized runoff index (SRI), standardized soil moisture index (SSI), and MSDI, gravity recovery and climate experiment (GRACE), correlation analysis | hydrological drought monitoring with longer record of GRACE observations. | DM | Green |
| [78] | West Africa | To examine the impacts of drought and responses of west African populations | systematic review of the literature | More research is needed on the efficiency and unanticipated effects of responses of populations, states, and NGOs, and interactions between different responses | DM | Green |
| [79] | West Africa | To assess the impacts of climate change and variability on drought characteristics | SPI, SPEI, standardized runoff index (SRI), | Studies on approaches to facilitate vulnerability assessment and adaptive capacity of the basin to minimize the negative effects of climate change. | DM | Green |
| [80] | Horn of Africa | To evaluate the use of European centre for medium-range weather forecasts (ECMWF) products in monitoring and forecasting drought conditions | era-interim reanalysis (ERAI), SPI, NDVI, ECMWF | The need for more global monitoring and forecasting of drought | DP | Gold |
| [81] | East Africa | To evaluate the use of ECMWF products in forecasting droughts | SPI, ECMWF | Further studies needed on meteorological seasonal forecast | DP | Gold |
| [82] | Southern Africa | To address the seasonal prediction of hydrological drought | ECMWF, ESP, conditional ESP approach (ESPCOND) | Further studies on hydrological drought seasonal forecast and skill assessment | DP | Gold |
| [83] | East Africa | To improve the characterization and quantification of vegetative drought as a ambiguous spatial phenomenon | NDVI, FUZZY Modelling | This method can also be used in other regions, otherwise adapted to characterize and quantify other vague spatial phenomena | DM | Yellow |
| [84] | South Africa | To analyze the vegetation response pattern of the oldest asserted nature reserve in Africa, | EVI, burned area index (BAI), and normalized difference infrared index (NDII), NDVI, statistical analysis | Further analytic studies on the vegetation response pattern to drought | DM | Yellow |
| [85] | South Africa | To assess the influence of drought on forest plantations | NDVI, NDII, palmer drought severity index (PDSI), statistical analysis | Further research on the forests' responses to drought is vital for management planning and monitoring. | DM | Yellow |
| [11] | Africa | To propose a novel method for calculating the empirical probability of having a substantial proportion of the entire agricultural area affected by drought at sub-national level. | VHI, NDVI | Future drought monitoring in Africa could be based on drought occurrence over both the temporal and the spatial domain | DM | Blue-grey |
| [86] | Kenya | To model temporal fluctuations of maize production and prices with a novel hyperspectral remote sensing method | NDVI, statistical analysis | Future research can consider adding other price-driving factors to the regression models. | DM | Blue-gray |
| [87] | South Africa | To analysis vegetation response to drought in diverse land management and land tenure systems | VCI, enhanced vegetation index (EVI), statistical analysis | Drought years can be detected by change in total annual vegetation productivity while drought dynamics during the season could be monitored by the VCI. | DM | Blue-gray |
| [88] | Ethiopia | To develop an experimental drought monitoring tool that predict the vegetation condition (vegetation outlook) | regression-tree technique, Vegout-Ethiopia, NDVI | Future studies are suggested that can help Eastern Africa in advancing knowledge of climate, remote sensing, hydrology, and water resources. | DP | Indigo |
| [89] | Ethiopia | to develop a remote sensing-based vegetation condition drought-monitoring approach for pastoralist areas | classification and regression tree (CART) modelling technique | Future research can improve both the administrative and spatial resolution of the model to determine drought status at district levels, useful for actual drought mitigation planning. | DP | Indigo |
| [90] | Kenya | To assess the performance of both heterogeneous and homogeneous model ensembles in the satellite-based prediction of drought severity | artificial neural networks (ANN), support vector regression (SVR), general additive model (GAM) technique | More research on the ex-ante drought early warning systems capable of offering drought forecasts with sufficient lead times | DP | Indigo |
| [91] | Ethiopia | To develop a higher-spatial-resolution vegetation outlook for upper Blue Nile (Vegout-UBN) model that is capable of integrating multiple satellite, climatic, and biophysical input variables | Vegout-UBN model, SPI and statistical analysis | The result can be used for potential application of Vegout-UBN for drought monitoring and prediction. | DP | Indigo |

**Note:** DM: Drought Monitoring, DP: Drought Prediction.

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
