# Peer review of "Bibliometric Analysis of Methods and Tools for Drought Monitoring and Prediction in Africa"

_sustainability, doi:10.3390/su12166516_

Round 1

Reviewer 1 Report

The revision of the article has lead to a clearly improved scientific systematic review. An author has been added but this seems not to cause problems. The title of the article has been reworded to a more accurate version although it is still somewhat vague: is the article a bibliometric analysis or a structural review, or both? The basics of the used methods have now been included and contents and written analysis of the reviewed articles presented to an appropriate level together with an appendix table. The discussion and conclusions now show reasoned linkages, analytical thinking and contribution of the article to the discipline.

One central deficiency of the article is its tight concentration on remote sensing. As the drought is a vast but apparently scientifically understudied environmental hazard in Africa, complementing data resources and records of knowledge should also be taken into account and described in the article, if not fully included in the review. At least, I suggest taking the term "remote sensing" to the main title.

Language in the article is mainly all right but the text contains too often careless writing and unpolished grammar. A careful word-by-word native proofreading is required. For example, among many others:
- L15 "Web of Science and Scopus database"
- L28 "contributed about"
- L71 "was were"
- L161 "African and 6 regions"
- L199 "Figure 2D": incorrect reference.
- L250 "Country collaborative analysis"
- L445 "monitoring and prediction early systems"
- L519 "drought studies green areas"

The article still contains many unclear points which necessitate revision, for instance:
- SPEI(-6) is central to the article. It must be written open already in the abstract and explicated in the first appearance in the text body. For example, how it must be interpreted in Figure 2 - why does it fall negative so sharply - is the graph built to show so?
- The authors should discuss at an appropriate point why they are interested in the correlation between the number of droughts and publications. The correlation itself is quite self-evident in the information era.
- L37: The opening sentence of the article is unnecessarily rhetorical. The article does not discuss the evolution process of the mankind.
- L52: Second time, "about 15 days" is not a few days.
- L109: the list of machine learning methods importantly lacks self-organising maps (SOM) by Kohonen.
- L134: "The use of WoS and Scopus in this review broadened the search" - compared to what?
- L150: Please cite appropriately when referring to the software and methods.
- L172: "excluding the northern regions": The authors must argue why they excluded these regions.
- Subsection 3.1.2: several insufficient or vague wordings to describe the analysis: "cluster property", "link property", "by the size of the circle", "strongest links", "strength association methodology". A scientific method must be repeatable by its description.
- L289 "Thematic evolution...": I think the authors mean rather development than evolution.
- Subsection 3.1.3: unclarity of description continues: "internal ties", "external ties", "concepts relevant to other themes".
- L521 "comprehensively": Do the authors think that the topics have been studied enough to say so?

When these shortcomings of the article are appropriately addressed, I think it is ready for publication.

Author Response

The authors sincerely thank the editor and reviewers for constructive criticism and recommendations provided for our manuscript. We have carefully addressed the reviewer’s valuable suggestions and comments and we now believe the manuscript is improved.

Point by point response is provided in the attached document

Reviewer 2 Report

I appreciate the changes Authors made. The paper is  better than when it first arrived and in my opinion can be published.

Author Response

The authors sincerely thank the editor and reviewer for constructive criticism and recommendations provided for our manuscript. 

Reviewer 3 Report

I am still firm in my previous decision. So, I can’t accept is as this manuscript does fulfil a review paper.

Author Response

The authors sincerely thank the editor and reviewers for constructive criticism and recommendations provided for our manuscript. 

This manuscript is a resubmission of an earlier submission. The following is a list of the peer review reports and author responses from that submission.

Round 1

Reviewer 1 Report

The review article entitled "Geospatial technologies and Machine Learning in advancing the knowledge of drought monitoring and prediction: A structural review" provides an overview on considerable number of research articles about drought research accounting the African continent. Essentially, the present article reports about use of common terms of drought research in these articles and assesses to which thematical collections these terms gather.

The topic of holistic and historical view on drought research is important and particularly so in the African continent where the vulnerability to lack of water is at the highest. However, the article misses a scientifically sound presentation of its topic. First, the title describes the topic inaccurately: focus on Africa is not mentioned and geospatial technologies as well as machine learning are very wide terms related to the contents of the article that do not discuss these technologies in detail. Second, the article tells almost nothing about the methods on which it builds its analyses. Most prominently, the reliability of the so-called "co-keyword network visualization", as interesting as the visualization is, is obscure as its making, as well as qualitative and quantitative interpretation remain unexplained. Comprehensive statistical numbers together with keyword visualizations are necessary for scientific robustness. Third, having read the article carefully through, its contribution to the scientific advancement needs improvement - the article reports use of terms and thereof rough topics of drought articles over time, but much more interesting contents, trends and phenomena can certainly be found by proper study of these articles. The found linkages of terms and thematic groupings seem random from outside of the analysis and authors do not provide enough arguments on their causes. Unfortunate as it is for its important topic, the article resembles more a rapidly collected report about figures from an automatic word-counting procedure, than a scientific journal publication.

Due to these reasons and following examples of loose writing, I would propose reject and resubmit, but as I see the potential of the topic and value of this research, I suggest a major revision with a possibility for reconsideration. However, major reformation and rewriting is absolutely necessary to make the article raise to the level of its topic in the proper scientific manner.

Examples of inaccuracies:

- L52 "a few days (about 15 days)": more accurate definition needed, please cite and elaborate
- L132 "reviews": whatdo you mean? Scientific review articles?
- bar figures: "Most Productive Countries": I think "productive" is a misleading term compared as these studies are not part of any pre-defined program
- L179: the last sentence is somehow malformed
- L187 and elsewhere "highest links": this important measure in the article is unclear and unexplained
- Figures 5 and 8: why do you divide the 2000s in two and what you reach by the division?
- L286 "human cognitive (indigenous knowledge)": unclear and unexplained sentence and wording
- L302 "59% of drought episodes occurred during the pre-2000 period": What do you mean? Certainly indefinite history has more events than 20 years.
- L303 "drought monitoring studies": what are these? Articles that contain certain terms? Not well-defined if so if contents are not studied deeper.
- L321 "vegetation index paramount among this is the NDVI": malformed sentence

Reviewer 2 Report

This study is a bibliometric review of studies on drought in Africa. I find that the bibliometric review itself is fairly done, but I also find that the authors need to argue some insights gained from the bibliometric review to stand out as a scientific paper. The current version does not contain any insights like the future direction of drought studies, so I cannot recommend this manuscript be considered publication.

Apart from the point above, I have following relatively minor points that should be addressed before being considered publication.

1. Figures 3 and 6 seem to concern with only the first or corresponding author. In other words, only one entry appears to be given per document. Then the questions are: 1) How did you deal with multi-national documents? 2) Did you pick up countries of the first author or the corresponding author? The first author is not always the corresponding author.

2. Drought monitoring studies are divided into four clusters (Figure 4). How can you justify the number of clusters? In other words, why is the number of cluster 4, not 3 or 5?

3. The caption of Figures 4 and 7 needs a fuller explanation. For example, the authors need to explain which color describes which, although it is described in the text. It is essential to organize figure captions so that readers can understand without reading the main text.

4. In Figures 4 and 7, what is the difference between panels A, B, and C?

5. How can you justify the number of clusters in Figure 7?

5. The authors need to define "Centrality" and "Density" in Figures 5 and 8.

6. Line 148 visualized outlined, and then...: visualized, outlined, and then... (with a comma)./

7. Line 175 Africa: African

Reviewer 3 Report

Drought is one of the most serious problems of the African continent. And drought research is very important to prevent its negative effects.

But I have a few doubts about this article.

Line 59-61: Therefore, studies on drought monitoring and prediction are essential to help stakeholders in making important decisions on water resources planning and management.

I fully agree with this statement, but what contribution the authors make to the development of drought research, besides searching and citing the research of other Authors?

Discussion section. Where the discussion is?

In my opinion Discussion section should showing the importance of your results and why your analyzes are important. What have you determined new in your research?

Line 326: Drought prediction studies in Africa were DISCOVERED to focus on four main themes, these are drought indicators, drought drivers, drought variables and remote sensing data. And line 335: The driving theme for drought monitoring studies in Africa is “El Niño”. And line 341: Drought monitoring and prediction studies were seen to have not only been conducted by African authors but by authors from other countries of the world. 

Is this your novelty in drought research? Very impressive.

Good Discussion section should contain what are the similarities and differences in your results concerning the drought research? Discussion is a kind of dialogue between your results and the results obtained by other researchers.

Did your results confirm the hypothesis? In my opinion Authors simply counted and sorted scientific papers concerning drought.

Otherwise, the manuscript is not well structured. Review publications do not have the IMRaD structure characteristic of original articles. The review publication does not present new research results, so it does not contain such parts as Materials, Methods, Results or Discussion. In my opinion (if it is to be a review article), the aim of the article should be e.g. a critical analysis of the diverse methods have been using for researche and predicting drought. 

Reviewer 4 Report

It is not well written, inadequately referenced and does not extend the existing knowledge in a meaningful way as a good review paper. While a review paper should draw useful conclusions, and provide guidelines to other researchers for future work broadly, the paper in question serves only as a list of (some) existing references. Though it does present some conclusions, they are not visibly supported by references and may have some author bias. One big area for improvement is providing more complete technical descriptions, here ML. More specifically, what classifier(s) was used, why used, what was the features/variables chosen, was it supervised/unsupervised etc., what was the results (e.g., accuracy, AOC, ROC), which cross validation methods used etc..

For all of these reasons, I must recommend the paper for rejection.